# FlowPG: Action-constrained Policy Gradient with Normalizing Flows

**Janaka Chathuranga Brahmanage, Jiajing Ling, Akshat Kumar**
School of Computing and Information Systems
Singapore Management University
{janakat.2022, jjling.2018}@phdcs.smu.edu.sg, akshatkumar@smu.edu.sg

## Abstract

Action-constrained reinforcement learning (ACRL) is a popular approach for solving safety-critical and resource-allocation related decision making problems. A major challenge in ACRL is to ensure agent taking a valid action satisfying constraints in each RL step. Commonly used approach of using a projection layer on top of the policy network requires solving an optimization program which can result in longer training time, slow convergence, and zero gradient problem. To address this, *first* we use a normalizing flow model to learn an invertible, differentiable mapping between the feasible action space and the support of a simple distribution on a latent variable, such as Gaussian. *Second*, learning the flow model requires sampling from the feasible action space, which is also challenging. We develop multiple methods, based on Hamiltonian Monte-Carlo and probabilistic sentential decision diagrams for such action sampling for convex and non-convex constraints. *Third*, we integrate the learned normalizing flow with the DDPG algorithm. By design, a well-trained normalizing flow will transform policy output into a valid action without requiring an optimization solver. Empirically, our approach results in significantly fewer constraint violations (upto an order-of-magnitude for several instances) and is multiple times faster on a variety of continuous control tasks.

## 1 Introduction

Action-constrained reinforcement learning (ACRL), where an agent's action taken in each RL step should satisfy specified constraints, has been applied to solve a diverse range of real-world applications [18]. One example is the resource allocation problem in supply-demand matching [4]. The allocation of resources must satisfy constraints such as total assigned resources should be within a specific upper and lower limit. In robotics, kinematics constraints (e.g., limits on velocity, acceleration, torque among others) can be modeled effectively using action constraints [1, 13, 26, 32].

One of the main challenges in solving ACRL is to ensure that during training and evaluation, all the action constraints are satisfied at each time step, while simultaneously improving the policy. Lagrangian Relaxation (LR) is a popular approach for solving constrained RL problems [30], however directly applying LR to ACRL is impractical due to the difficulty in defining cost functions $(c(s, a))$ that penalize infeasible actions for all $(s, a)$, and the inability to guarantee zero constraint violation during training and policy execution. Also, there are studies addressing ACRL with discrete action spaces [19, 20]. As they use propositional logic, it is not clear how to extend them to continuous action spaces. Recently proposed approaches aim to satisfy continuous action constraints using an *action projection* step. One natural approach is to add a differentiable projection layer at the end of the policy network to satisfy the constraints [2, 4, 7, 26]. This projection layer projects the unconstrained policy action onto the feasible action space by solving a quadratic program (QP). However, this approach has two primary limitations. Firstly, solving a QP at each RL step can be

37th Conference on Neural Information Processing Systems (NeurIPS 2023).

computationally expensive, particularly when dealing with non-convex action constraints or large action spaces. Secondly, the coupling between the policy network and projection layer can lead to a potential issue of zero gradient during end-to-end training, which can undermine the effectiveness of the projection layer approach, especially in the early training stages, as highlighted by recent studies [18]. This issue occurs when the policy output is far outside the feasible action space, thereby any small change in the policy parameters does not result in any change in the projected action.

To overcome the zero gradient issue, an ACRL algorithm has been proposed in [18], which decouples policy gradients from the projection layer. The policy parameters are updated by following the direction found by the Frank-Wolf method [10], which is within the feasible action space. However, this approach can also be computationally expensive, as solving a QP is still required during the update of policy parameters and during policy execution. To summarize, while there are several proposed approaches for solving ACRL with continuous action space, all of them share the common drawback of requiring QP solving during either training or action execution, or both. This can cause a significant reduction in training speed, which we also validate empirically. Moreover, some of these approaches also suffer from the zero gradient issue, which can make training sample inefficient. Therefore, there is need for more efficient methods to tackle ACRL with continuous action spaces.

Often, action constraints are specified analytically using features from state and actions (e.g., $x^2 + y^2 <= 1$ where $x, y$ can be features). Our key idea is to exploit given action constraints as domain knowledge and develop an effective, compact representation of all valid actions that can be integrated with RL algorithms. To find such a representation, generative models provide an attractive solution since they can generate valid actions by learning from a finite number of valid actions (which can be provided by sampling from the valid action space). This allows for a compact feasible action space representation even for continuous action spaces. Furthermore, generative models can be easily integrated with RL algorithms for end-to-end training, as shown in previous work [22, 33], even though they are not specifically solving ACRL problems. The idea of learning the representation of all valid actions also shares a connection with offline RL, where invalid actions are often treated as out-of-distribution (OOD) actions [12, 35]. However, a notable difference between our work and offline RL lies in the availability of a dataset comprising valid actions. In offline RL, such a dataset is assumed to be available (e.g., collected from a random policy or an expert). In ACRL, collecting data even from a random policy is not straightforward as random policy must also select actions uniformly from the feasible action space, which itself is the key problem we address in this work.

**Our main contributions are:**

- We utilize the normalizing flows, a type of generative model, to learn a differentiable, invertible mapping between the data distribution of sampled valid actions and a simple latent distribution (e.g., Gaussian, uniform distribution) [9, 27]. When a member of the latent distribution is given, the flow model can transform it into an element that conforms to the distribution of valid actions, assuming the model is well-trained. Compared to other generative models, such as Generative Adversarial Networks (GANs) and Variational Autoencoders (VAEs), the normalizing flow model is known to be more efficient at completing the task of data generation [9].

- Sampling uniformly from the feasible action space to train the flow model is challenging. For example, valid action space may be characterized using a set of constraints over binary variables. We therefore develop multiple methods, based on Hamiltonian Monte-Carlo [3] and probabilistic decision diagrams (PSDD) [16, 28], for valid action sampling for (non)convex constraints. These methods are significantly more sample efficient than the standard rejection sampling.

- We propose a simple and easy to implement method to integrate the normalizing flow model with deep RL algorithms such as DDPG [17]. Our approach is general, requires change only in the last layer of the policy network, and can be integrated with other deep RL methods. In our method, the modified policy network outputs an element of the latent distribution used in the normalizing flow model, which is then transformed into a valid action via the mapping provided by the normalizing flow. We also show gradients can be propagated through the normalizing flow to improve the policy. By integrating the normalizing flow with deep RL algorithms, we can avoid the zero gradient issue since there is no projection of invalid action onto the feasible action space. We show empirically that the flow model can be trained well on a variety of action constraints used in the literature, therefore, probability of constraint violation occurring remains low. Empirically, our approach results in significantly fewer constraint violations (upto 10x less), and is multiple times faster on a variety of continuous control tasks than the previous best method [18].

## 2 Preliminaries

**Action-constrained Markov Decision Process** We consider a Markov decision process (MDP) model, which is defined by a tuple $(S, A, p, r, \gamma, b_0)$, where $S$ is the set of possible states, $A$ is the set of possible actions that an agent can take, $p(s_{t+1}|s_t, a_t)$ is the probability distribution over the next state $s_{t+1}$ given the current state $s_t$ and action $a_t$, $r(s_t, a_t)$ is the reward function that determines the reward received by the agent after taking action $a$ in state $s$ at time step $t$, $\gamma \in [0, 1)$ is the discount factor, and $b_0$ is the initial state distribution. We assume the action space $A$ is continuous, and focus on deterministic policies. Specifically, let $\mu_\theta(\cdot)$ denote a deterministic policy parameterized by $\theta$; action taken in state $s$ is given as $a = \mu_\theta(s)$. Under policy $\mu_\theta$, the state value function $(V)$ and the state-action value function $(Q)$ are defined as follows respectively.

$$V^{\mu_\theta}(s) = \mathbb{E}[\sum_{t=0}^\infty \gamma^t r(s_t, a_t)|s_0 = s; \mu_\theta], \ Q^{\mu_\theta}(s, a) = \mathbb{E}[\sum_{t=0}^\infty \gamma^t r(s_t, a_t)|s_0 = s, a_0 = a; \mu_\theta] \quad (1)$$

An action-constrained MDP extends the standard MDP by incorporating explicit action constraints that determine a valid action set $\mathcal{C}(s) \subseteq A$ for each state $s \in S$. In other words, the agent can only choose an action from the set of valid actions at each time step. The goal of an action-constrained MDP is to find a policy that maximizes the expected discounted total reward while ensuring that all chosen actions are valid with respect to the constraints. Formally, we have:

$$\max_\theta J(\mu_\theta) = \mathbb{E}_{s \sim b_0}[V^{\mu_\theta}(s)] \quad \text{s.t.} \ a_t \in \mathcal{C}(s_t) \quad \forall t \quad (2)$$

We conisder a RL setting where transition and reward functions are not known. The agent learns a policy by interacting with the environment and using collected experiences $(s_t, a_t, r_t, s_{t+1})$.

**Deep Deterministic Policy Gradient** Deep Deterministic Policy Gradient (DDPG) is a RL algorithm specifically designed to handle continuous action spaces [17]. The algorithm combines deterministic policy gradient [29] with deep $Q$-learning [25]. To solve an *unconstrained* RL problem, DDPG applies stochastic gradient ascent to update policy parameters $\theta$ in the direction of the gradient $\nabla_\theta J(\mu_\theta)$. The deterministic policy gradient theorem is used to compute the policy gradient as:

$$\nabla_\theta J(\mu_\theta) = \mathbb{E}_{s \sim \mathcal{B}}[\nabla_a Q^{\mu_\theta}(s, a; \phi)\nabla_\theta \mu_\theta(s)|_{a=\mu_\theta(s)}] \quad (3)$$

The state-action value function $Q(s, a; \phi)$ is approximated using a deep neural network, and the parameters $\phi$ of this neural network are updated by minimizing the following loss function:

$$\mathbb{E}_{(s,a,s',r) \sim \mathcal{B}}[(r + \gamma Q(s', \mu_{\theta'}(s'); \phi') - Q(s, a; \phi))^2] \quad (4)$$

where $(s, a, s', r)$ is a transition sample; $\mathcal{B}$ is the replay buffer that stores transition samples; $\theta'$ and $\phi'$ are the parameters of the target policy and the target state-action value function respectively.

**Normalizing Flows** A normalizing flow model is a type of generative model that transforms a simple distribution, such as a Gaussian or uniform distribution, into a complex one by applying a sequence of invertible transformation functions [9, 27]. Let $f = f_K \circ f_{K-1} \circ \ldots \circ f_1 : \mathbb{R}^D \mapsto \mathbb{R}^D$ denote a normalizing flow model, where each $f_i : \mathbb{R}^D \mapsto \mathbb{R}^D$, $i = 1 : K$, is an invertible *transformation* function. Starting from an initial sample from a base distribution (or prior distribution), $z_0 \sim p(z^0)$, the transformed sample from the model is $x = f_K \circ f_{K-1} \circ \ldots \circ f_1(z^0)$. Each $f_i$ takes input $z^{i-1}$ and outputs $z^i$, and $x = z^K$. Given a training dataset $\mathcal{D}$, the mapping function $f$ is learned by maximizing the log-likelihood of the data, which is defined as $\log p(\mathcal{D}) = \sum_{x \in \mathcal{D}} \log p(x)$. The log probability $\log p(x)$ is computed by repeatedly applying the change of variables theorem, and is expressed as:

$$\log p(x) = \log p(z^0) - \sum_{i=1}^K \log \left| \det \frac{\partial f_i(z^{i-1})}{\partial (z^{i-1})^T} \right| \quad (5)$$

## 3 Normalizing Flow for ACRL

### 3.1 Learning Valid Action Mapping Using Normalizing Flows

We employ normalizing flows to establish an invertible mapping between the support of a simple distribution and the space of valid actions. While various transformation functions can be utilized

for implementing normalizing flows, we specifically focus on the conditional RealNVP model [34] due to its suitability for the general ACRL setting, where the set of valid actions is dependent on the state variable. The conditional RealNVP extends the original RealNVP [9] by incorporating the conditioning variable in both the prior distribution and the transformation functions. These transformation functions, which are implemented as affine coupling layers, possess the advantageous properties of enabling efficient forward propagation during model learning and efficient backward propagation for sample generation [9].

Given a dataset $\mathcal{D}$ consisting of valid actions (details in Section 3.2), we train the invertible mapping $f_\psi$ parameterized by $\psi$ to capture the relationship between the support of a uniform distribution $p$ and the elements in $\mathcal{D}$. Learning this mapping provides an easy way to generate new valid actions. The policy network outputs an element from the domain of the uniform distribution, which is then mapped to a valid action using the learned flow. This process of generating actions is much simpler than using a math program for action projection, which can result in the zero gradient issue.

The flow learning process involves maximizing the log-likelihood of the data, following the methodology presented in [34]. In contrast to using a Gaussian distribution, we opt for a uniform distribution as it demonstrated better empirical performance when combined with the DDPG algorithm. Once the normalizing flow model is learned, during backward propagation with conditioning variables, each bijective function $f_{\psi_i}$, where $i = 1, \ldots, K$, takes inputs of $z^{i-1} \in \mathbb{R}^D$ and a conditioning variable $y \in \mathbb{R}^{D_y}$ (representing state features) and produces an output of $x^i \in \mathbb{R}^D$ as:

$$x^i_{1:d} = z^{i-1}_{1:d} \tag{6}$$
$$x^i_{d+1:D} = (z^{i-1}_{d+1:D} - t(z^{i-1}_{1:d}, y)) \odot \exp(-k(z^{i-1}_{1:d}, y)) \tag{7}$$

where $d < D$ is the index that splits dimensions of input $x^i$ into two parts; $k$ and $t$ are scale and translation functions that map from $\mathbb{R}^{d+D_y} \mapsto \mathbb{R}^{D-d}$. These two functions are modeled using neural networks. $\odot$ denotes the element-wise product (Hadamard product). Therefore, given a random member of the uniform distribution $z^0 \sim p(z^0)$ and a conditioning variable $y$, the mapped element is:

$$x = f_{\psi_K} \circ f_{\psi_{K-1}} \circ \ldots \circ f_{\psi_1}(z^0, y) \tag{8}$$

Note that the transformed output $x$ obtained through the normalizing flow model may not precisely match any specific element from the dataset $\mathcal{D}$. However, when the mapping $f_\psi$ is effectively trained, the resulting $x$ should exhibit similar characteristics to the elements in $\mathcal{D}$, thereby satisfying the constraints with high probability, as we also observed empirically. In essence, the combination of the uniform distribution and the learned mapping serves as an efficient representation of all valid actions, providing a means to generate actions that adhere to the constraints.

Unlike other generative models, normalizing flows provide the ability to measure the recall rate in learning the valid action mapping due to their invertible bijective transformations. The recall rate (we call it "coverage") indicates the fraction of valid actions that can be generated from the latent space. Given a conditioning variable $y$, let $\mathcal{C}(y)$ denote the set of valid actions that are uniformly distributed in the feasible region. The recall can be computed as follows.

$$recall(y) = \frac{\sum_{x \in \mathcal{C}(y)} \mathbb{I}_{dom_{f_\psi}} f_\psi^{-1}(x, y)}{|\mathcal{C}(y)|} \tag{9}$$

where $f_\psi^{-1}$ is the inverse transformation function of the normalizing flows model $f_\psi$. Recall is useful to characterize the learned generative model. If the recall is low, it would limit the feasible constrained space over which ACRL algorithm optimizes the policy. This can result in lower solution quality.

**The Mollified Uniform Prior**    Most recent studies of normalizing flows employ a standard Gaussian distribution as the latent space distribution [9, 15, 27], which spans the entire real number space $\mathbb{R}^D$. However, through empirical observation, we noticed that with Gaussian latent distribution, the feasible action region tends to be mapped *closer to zero* in the latent space, where the probability of the standard Gaussian is higher. On the other hand, points located far away from the center of the Gaussian (several standard deviations) often get mapped into the *infeasible* action space. Consequently, when coupling the DDPG policy network with the flow, it resulted in high constraint violations as the policy network output can be any number from the support of the Gaussian, and not just limited to the high probability region.

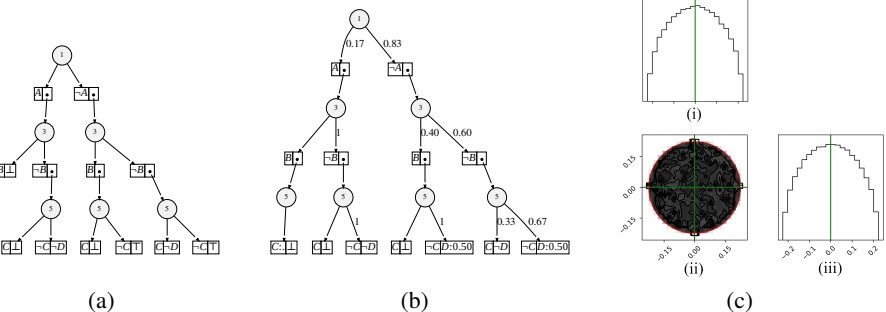

| (a) | (b) | (c) |

Figure 1: (a)An SDD representing the PB constraint $A \cdot 2^1 + B \cdot 2^0 + C \cdot 2^1 + D \cdot 2^0 \leq 2$; (b)A PSDD ; (c) Samples generated with HMC

To address this issue, we decided to use a uniform distribution from $[-1, 1]^D$ as the latent distribution instead of the standard Gaussian. In the uniform distribution, there is no high or low probability region unlike the standard Gaussian. However, the uniform distribution is not differentiable, therefore we employ a modified version called the Mollified uniform distribution [11]. In order to train the normalizing flow, we require the probability density of the prior distribution as in Eq. 5. Computing the probability density of a multi-dimensional mollified distribution can be challenging since it requires integrating over all dimensions [11]. However, in the case of a single dimension, a sample drawn from a mollified uniform distribution can be seen as a sample from a uniform distribution with added Gaussian noise. In other words, a sample $x$ is obtained as $x = x_0 + \delta$, where $x_0$ follows a uniform distribution in the range $[-1, 1]$, and $\delta$ follows a Gaussian distribution with mean 0 and standard deviation $\sigma$.

To calculate the overall probability density, we treat each dimension individually, compute their probability densities, and take their product as the overall probability. The probability density in a single dimension can be easily calculated using the following formula: $p(x) = \int_{-1}^{1} \frac{1}{\sqrt{2\pi\sigma^2}} e^{-\frac{1}{2\sigma^2}(x-x_0)^2} dx_0 = \Phi\left(\frac{1-x}{\sigma}\right) - \Phi\left(\frac{-1-x}{\sigma}\right)$. In this formula, $\Phi$ represents the cumulative density function of the standard Gaussian distribution.

### 3.2 Generating Random Samples from the Valid Action Space

To effectively train the normalizing flow model, we need to generate samples from the set of valid actions. Even with the known analytical form of action constraints this is a challenging problem. Standard rejection sampling is highly inefficient to sample from the complex constraints that arise in ACRL benchmarks. Therefore, we use two methods described next.

**Hamiltonian Monte-Carlo (HMC)** HMC is a Markov chain Monte Carlo method, which utilizes energy conservation to effectively sample a target distribution [3]. It utilizes Hamiltonian-dynamics to simulate the movement of a particle in a continues space based on a potential energy function. It uses leap-frog algorithm with the first order integrator, to estimate the next position of the particle. In our work, we need to generate samples in the feasible region. Therefore we define energy function to be a fixed value 0 in the feasible region. Then we bound the feasible region with a hard wall of infinity potential by setting the potential to $\infty$ in the infeasible region. Figure 1(c) depicts the distribution of feasible action samples generated by HMC in the action constraint of the Reacher Environment $(a_1^2 + a_2^2 \leq 0.05)$. It presents a contour map of the sample density in Figure 1(c)-(ii), as well as their projection onto each dimension in Figure 1(c)-(i) and Figure 1(c)-(iii) respectively.

**Probabilistic Sentential Decision Diagrams** In many real-world applications such as resource allocation, the constraints are defined using a set of linear equalities/inequalities over integer variables. However, sampling valid actions using HMC that satisfy such constraints is very challenging due to the presence of equality constraints. To address this challenge, we propose to use a Probabilistic Sentential Decision Diagram (PSDD) [16] to encode a uniform distribution over all the valid actions and then sample actions from the PSDD. For this setting, we assume that all the constraints are linear (in)equalities. We describe the process of constructing a PSDD next. Consider an inequality constraint $x + y \leq 2$. We follow the following steps:

1. We convert the variables into binary representations. The number of bits used in the binary representation depends on the upper/lower bounds of variables. For example, assuming $x, y \in \{0, 1, 2, 3\}$, we can use $A \cdot 2^1 + B \cdot 2^0$ to represent $x$ and $C \cdot 2^1 + D \cdot 2^0$ to represent $y$ where $A, B, C, D$ are Boolean variables.

2. We convert the given linear constraint into a Pseudo-Boolean (PB) constraint by using the binary representation of variables. The resulting PB constraint is $A \cdot 2^1 + B \cdot 2^0 + C \cdot 2^1 + D \cdot 2^0 \leq 2$. Note that the coefficients in a PB constraint can be real numbers.

3. We compile the PB constraint into an SDD using the method described in [6]. An SDD encodes all instantiations of Boolean variables that satisfy the constraint. Each valid instantiation is called a *model* of the SDD. Figure 1(a) shows a compiled SDD for the PB constraint. For the syntax and semantics of SDD, we refer to [8] for detailed elaboration. When there are multiple equalities/inequalities constraints, we create SDDs in this fashion for each constraint. We then conjoin all such SDDs (which is a poly-time operation) to encode all the constraints into a single SDD that represents the valid action space satisfying all the constraints.

4. We parameterize the final SDD using a standard package [5] to obtain a PSDD as shown in Figure 1(b), which represents a probability distribution over all models of the underlying SDD. A uniform distribution over all models is achievable by a special parameter initialization scheme as provided in [5]. Sampling models from the PSDD can be efficiently performed using a fast and top-down procedure, as described in [16].

We use this scheme to sample valid resource allocation actions in a bike sharing domain that has multiple equality and inequality constraints and has been used previously [4, 18].

## 3.3 Integrating DDPG with Normalizing Flows

In previous approaches to ACRL, addressing zero constraint violations often involves adding a projection layer to the original policy network. However, this method has drawbacks such as increased training time, slow convergence, and the issue of zero gradient when updating policy parameters, particularly during the early stages of training when pre-projection actions are likely to be invalid. In our work, we propose an integration of the DDPG algorithm with the *learned* mapping between the support of a uniform distribution $p$ and the set of valid actions as in Section 3.1. This integration allows us to incorporate the learned mapping directly into the original policy network of DDPG, alleviating the aforementioned issues.

Figure 2(a) illustrates the architecture of our proposed policy network, which consists of two modules.

- The first module is the original parameterized policy network $\mu_\theta$ from DDPG, which takes the state $s$ as input and outputs $\tilde{a}$. To ensure that $\tilde{a}$ belongs to a uniform distribution $p$ used for learning the mapping $f_\psi$, we employ a Tanh activation function at the end of the original policy network.

- The second module is the learned mapping function $f_\psi$ using normalizing flows. This mapping function takes inputs of $\tilde{a}$ and $s$.

Since $\tilde{a}$ is a member of the uniform distribution $p$, for a well trained $f_\psi$, chances of the mapped action outside of the valid action space will be low, which we also observe empirically. Our proposed policy network offers two significant advantages compared to previous ACRL methods. First, it eliminates the need to solve a QP to achieve zero constraint violation. Instead, the valid action is obtained by mapping a sample from the uniform distribution, resulting in a substantial increase in training speed. Note that in our work the mapping function $f_\psi$ is pre-trained, and its parameters $\psi$ remain fixed during the learning process of the original policy parameters $\theta$. However, if it becomes necessary for large state/action spaces, we can adapt our approach to refine the flow during the learning process. This can be achieved by performing a gradient update on $f_\psi$, using newly encountered states and valid actions, as described in Section 3.1. Through our experiments, we demonstrate the accuracy of the learned mapping, highlighting the effectiveness of our approach to satisfy action constraints. Second, the architecture of our policy network enables end-to-end updating of the original policy parameters $\theta$ without solving a math program for action projection, thereby avoiding the issue of zero gradient. This advantage allows for smoother training and more stable convergence.

**Policy update** The objective is to learn a deterministic policy $f_\psi(\mu_\theta(s), s)$ that gives the action $a$ given a state $s$ such that $J(\mu_\theta)$ is maximized. We assume that the $Q$-function is differentiable with respect to the action. Given that the learned mapping $f_\psi$ is also differentiable and the parameters $\psi$ are frozen, we can update the policy by performing gradient ascent only with respect to the original policy network parameters $\theta$ to solve the following optimization problem:

$$\max_\theta J(\mu_\theta) = \mathbb{E}_{s\sim\mathcal{B}}[Q(s, f_\psi(\mu_\theta(s), s); \phi)] \tag{10}$$

where $\mathcal{B}$ is the replay buffer and $Q$-function parameters $\phi$ are treated as constants. Figure 2(b) shows the reversed gradient backpropagation path (in blue) for $\theta$. The new deterministic policy gradient for the update of $\theta$ is then given as follows.

$$\nabla_\theta J(\mu_\theta) = \mathbb{E}_{s\sim\mathcal{B}}[\nabla_a Q^{\mu_\theta}(s, a; \phi)\nabla_{\tilde{a}} f_\psi(\tilde{a}, s)\nabla_\theta \mu_\theta(s)|_{\tilde{a}=\mu_\theta(s), a=f_\psi(\tilde{a}, s)}] \tag{11}$$

Next, we derive the analytical form of the gradient term $\nabla_{\tilde{a}} f_\psi(\tilde{a}, s)$ in the new deterministic policy gradient, which is an additional component compared to the standard DDPG update (3). For notational simplicity, we focus on computing $\nabla_{z^0} f_\psi(z^0, y)$, which is same as $\nabla_{\tilde{a}} f_\psi(\tilde{a}, s)$ (as last layer of the policy maps to the domain of the latent uniform distribution, or $\tilde{a} = z^0$). We also note $\psi$ is the collection of parameters $\psi_i \, \forall i = 1, \dots, K$. To compute this gradient, we apply the chain rule of differentiation. Considering the composition of functions $f_\psi(z^0, y) = f_{\psi_K}(f_{\psi_{K-1}}(\dots f_{\psi_1}(z^0, y)))$, we can express the derivative as a product of gradients:

$$\nabla_{z^0} f_\psi(z^0, y) = \frac{\partial f_{\psi_K} \circ f_{\psi_{K-1}} \circ \dots \circ f_{\psi_1}(z^0, y)}{\partial(z^0)^T}(z^0)$$

$$= \frac{\partial f_{\psi_1}}{\partial(z^0)^T}(z^0) \cdot \dots \cdot \frac{\partial f_{\psi_K}}{\partial(z^{K-1})^T}(z^{K-1} = f_{\psi_{K-1}}(z^{K-2}, y)) \tag{12}$$

The Jacobian of each bijective function $f_{\psi_i}, i = 1, \dots, K$ is

$$\frac{\partial f_{\psi_i}}{\partial(z^{i-1})^T} = \frac{\partial x^i}{\partial(z^{i-1})^T} = \begin{bmatrix} \mathbb{I}_d & \mathbf{0} \\ \frac{\partial x^i_{d+1:D}}{\partial(z^{i-1}_{1:d})^T} & \text{diag}(\exp[-k(z^{i-1}_{1:d}, y)]) \end{bmatrix} \tag{13}$$

where $\mathbb{I}_d$ is a $d \times d$ identity matrix. $\text{diag}(\exp[-k(z^{i-1}_{1:d}, y)])$ is the diagonal matrix where the diagonal consist of elements corresponding to the vector $\exp[-k(z^{i-1}_{1:d}, y)]$ and the other elements are all zeros. $\partial x^i_{d+1:D}/\partial(z^{i-1}_{1:d})^T$ is computed as follows.

$$\frac{\partial x^i_{d+1:D}}{\partial(z^{i-1}_{1:d})^T} = -\exp[-k(z^{i-1}_{1:d}, y)]\mathbf{1}^T \odot \left( [z^{i-1}_{d+1:D} - t(z^{i-1}_{1:d}, y)]\mathbf{1}^T \odot \frac{\partial k(z^{i-1}_{1:d}, y)}{\partial(z^{i-1}_{1:d})^T} + \frac{\partial t(z^{i-1}_{1:d}, y)}{\partial(z^{i-1}_{1:d})^T} \right) \tag{14}$$

where $\mathbf{1}^T$ denotes a row vector of length $d$ whose elements are all one. $\frac{\partial k(z^{i-1}_{1:d}, y)}{\partial(z^{i-1}_{1:d})^T}$ and $\frac{\partial t(z^{i-1}_{1:d}, y)}{\partial(z^{i-1}_{1:d})^T}$ are the Jacobian of $k$ and $t$ respectively. Note that when $k$ and $t$ are complex functions modeled by neural networks. The Jacobian matrix can be computed efficiently using automatic differentiation tools such as Pytorch and Tensorflow.

**Critic Update** The update of $Q$-function parameters $\phi$ in our approach follows the same update rule as in DDPG (4). However, in case the mapped action by the learned flow does not satisfy action constraints, we solve a QP to project it into the feasible action space as the environment only accepts valid actions. Therefore, the action stored in the replay buffer $\mathcal{B}$ is either the output from the flow model or the projected action, depending on whether flow mapped action is valid or not. We note that the probability of using the projected action during training is low as for all the tested instances as the normalizing flow had high accuracy. Even if the projection was required, the difference between the flow output and the projected action was small as we show empirically. We also highlight that for policy update, we use the flow mapped action without using any projection, which enables end-to-end training of the policy.

**Other RL algorithms** While our work focuses on DDPG, it can be extended to other RL algorithms such as SAC or PPO. This is possible because the normalizing flows enable the computation of log probabilities of actions, which is required during training in SAC or PPO. This showcases an additional advantage of using the normalizing flow model in our work compared to other generative models.

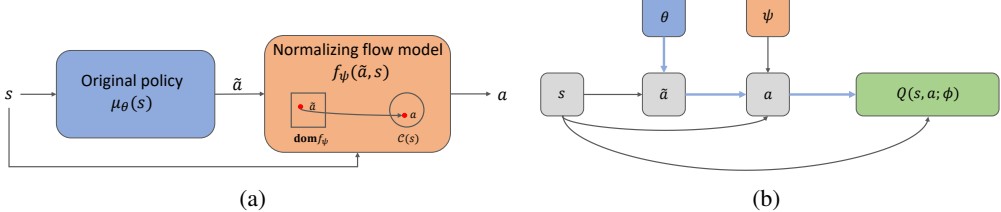

(a)              (b)

Figure 2: (a) Policy network; (b) The reversed gradient path of $\theta$ (in blue). Nodes denote variables and edges denote operations. Paths in black are detached for $\theta$. The green block is a negative loss (assuming a minimization task).

## 4 Experiments

**Environments** We evaluate our proposed approach on four continuous control RL tasks in the MuJoCo environment [31] and one resource allocation problem that has been used in previous works [4, 18].

- **Reacher:** The agent is required to navigate a 2-DoF robot arm to reach a target. The agent's actions $(a_1, a_2) \in \mathbb{R}^2$ are 2D. The action space is subject to a nonlinear convex constraint $a_1^2 + a_2^2 \leq 0.05$.
- **Half Cheetah:** The task is to make the cheetah run forward by applying torque on the joints. An action is a 6-dimensional vector $(v_1, v_2, ...v_6)$, bounded by $[-1, 1]$. The constraint is defined as $\sum_{i=1}^{6} |v_i w_i| \leq 20$ where $w_i$ is the velocity, which can be observed as part of the state.
- **Hopper and Walker2d:** The task involves controlling a robot to hop or walk forward by applying torques to its hinges. An action is a $n$-dimensional vector $(v_1, ..., v_n)$, bounded by $[-1, 1]$ where $n$ is the number of hinges on the robot (3 for Hopper and 6 for Walker2d). The constraint is defined as $\sum_{i=1}^{n} \max\{w_i v_i, 0\} \leq 10$ where $w_i$ is the angular velocity of the $i^{th}$ hinge, observed in the state.
- **Bike Sharing System (BSS):** The environment consists of $m$ bikes and $n$ stations, each with a capacity $c$. At every RL step, the agent allocates $m$ bikes among $n$ stations based on previous allocation and demand while adhering to capacity constraints. The agent's action $a = (a_1, a_2, ..a_n)$ must satisfy $\sum_{i=1}^{n} a_i = m$ and $0 \leq a_i \leq c$. We evaluate our approach on a specific scenario with $n = 5$, $m = 150$, and $c = 35$ as in [18]. It poses a significant challenge due to the large combinatorial action space of $151^5$.

**Baselines** We compare our approach FlowPG with the following two baselines.

- **DDPG+P:** DDPG+Projection is an extension of the vanilla DDPG [17], which introduces an additional step to ensure feasible actions. Invalid actions are projected onto the feasible action space by solving an optimization problem.
- **NFWPO [18] :** This algorithm efficiently explores feasible action space by using the Frank-Wolf method, and is also state-of-the-art approach for ACRL.

**Learning the Normalizing Flows** We employ different techniques to generate random samples from the valid action set in each environment. For the Reacher and Half Cheetah environments, we utilize HMC to generate random samples. Specifically, we generate 1 million random samples for training, which could take up to 5 minutes based on the constraint. HMC is more efficient than rejection sampling and a comparison is available in Appendix A. In the Reacher and Half Cheetah environments, all samples generated by HMC are valid actions, which shows the effectiveness of HMC compared to traditional rejection sampling which only produces 3.93%, 4.70% valid actions respectively. For the BSS environment, we use a PSDD to sample valid actions and obtain all valid actions from the PSDD to train the flow. Additional information on compiling the linear constraints into a PSDD and statistics about the PSDD are provided in the Appendix A.

We apply batch gradient descent to train the conditional flow with Adam optimizer and a batch size of 5000. For Reacher, Half Cheetah, Hopper and Walker2d environments, we train the model for 5K epochs. For BSS environment, we train for 20K epochs. Further details about the training the flow such as learning rates, and neural network architecture of the model are provided in the Appendix B.

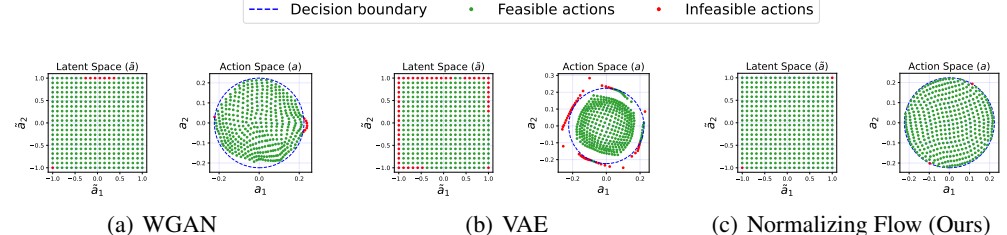

(a) WGAN        (b) VAE       (c) Normalizing Flow (Ours)

Figure 3: Mapping between a uniform distribution and action space of Reacher with constraint $a_1^2 + a_2^2 \leq 0.05$

The source code of our implementation is publicly available[1]. To evaluate the accuracy of the flow model, we sample 100K random points from the uniform distribution $[-1, 1]$, and then apply the flow. We measure the accuracy based on whether the output lies within the feasible action region. Our flow model was able to produce 99.98%, 97.25%, 87.89%, 86.58% and 85.56% accuracy respectively for Reacher, Half Cheetah, Hopper, Walker2d and BSS environments. Further, we compute the average recall over all uniformly sampled states to obtain the recall of our flow model using Equation 9. The achieved recall rates for our trained normalizing flow models were 97.85%, 78.01%, 81.61%, 83.58%, and 82.35% respectively for Reacher, Half Cheetah, Hopper, Walker2d and BSS environments. More details such as time are available in Table 1 in Appendix B.

**Comparing with other generative models** We conduct an ablation study comparing two other generative models VAE and WGAN (which is more stable than GAN) in Reacher domain. We first evaluated the accuracy by calculating the percentage of valid actions among 100k generated actions. The accuracy rates were as follows: Normalizing flows: 99.98%; WGAN 98%; VAE: 83%. We then considered the recall rate. Our flow model can achieve a recall rate of 97.85%. In contrast, recall rate cannot be computed in a straightforward fashion in VAE and WGAN since determining the corresponding latent action for a given valid action is not possible [21]. Nonetheless, we still can visualize the coverage of VAE and WGAN. In Figure 3, we can see that the feasible region is not fully covered in both VAE and WGAN models, while it is well covered in the normalizing flow model.

**Results** We present the empirical results of our approach and the baselines on three different environments to show the effectiveness of our method in terms of low constraint violations and achieving fast training speeds. We report results using ten different random seeds. To make a fair comparison, we keep the architectures of our policy network (with the exception of the Tanh layer and the normalizing flow model) and the critic network identical to those used in the baseline approaches. For the detailed pseudo-code of our approach and the neural network architectures along with hyperparameters, please refer to the appendix.

Figure 4(a) shows the average return of all three approaches over the training steps. Our approach achieves a comparable average return to NFWPO in the challenging Half Cheetah, Hopper and Walker2d environments. Additionally, our approach outperforms the other two approaches in terms of average return in the Reacher and BSS environments. We note that DDPG+P suffers from the zero gradient issue, leading to convergence to a lower average return.

In Figure 4(b), we present the cumulative constraint violations before projection for all three approaches over the training steps, with the y-axis represented in log-scale. In all environments, except for the more challenging Walker2d, our approach demonstrates the fewest cumulative constraint violations before projection. In the Reacher and Half Cheetah environments, it notably reduces the violations by an order of magnitude. Moreover, although our approach sometimes generates infeasible actions, they tend to be located near the feasible region, which can be inferred from the average magnitude of constraint violations. To quantify the magnitude of constraint violations, we consider a constrained set $\mathcal{C}$ defined with $m$ inequality constraints and $n$ equality constraints: $\{x | x \in \mathbb{R}^d, f_i(x) \leq 0, h_j(x) = 0, i = 1, \ldots, m, j = 1, \ldots, n\}$. We define the magnitude of constraint violations as $CV(x) = \sum_{i=1}^{m} \max(f_i(x), 0) + \sum_{j=1}^{n} \max(||h_j(x)| - \epsilon|, 0)$, where $\epsilon = 0.1$ represents the error margin for the equality constraint. Figure 4(c) illustrates the average magnitude of constraint violations over training steps, with the y-axis presented in log-scale. Our approach exhibits the lowest average magnitude of constraint violations in all environments except Hopper and

---

[1]https://github.com/rlr-smu/flow-pg

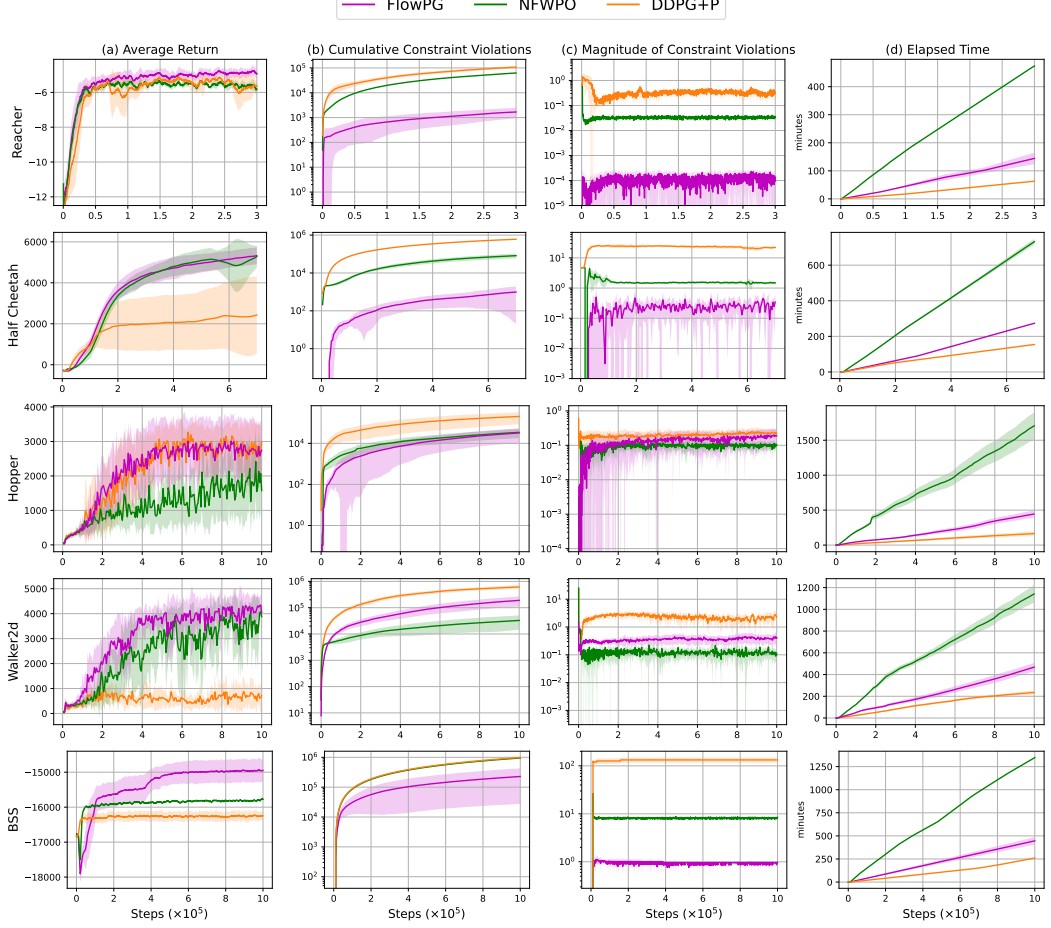

Figure 4: Training curves for the Reacher, Half Cheetah, and BSS environments are displayed in columns from left to right, showcasing the Average Return(↑), Cumulative Constraint Violations(↓), Average Magnitude of Constraint Violations(↓), and Time Elapsed(↓).

Walker2d. In the Reacher, Half Cheetah, Hopper and Walker2d environments, the average magnitude of constraint violations is close to zero, indicating that the invalid actions are very close to the feasible region. In the BSS environment, our approach shows slightly higher average magnitude due to the integer property of the actions. The low number of cumulative constraint violations and lower average magnitude of constraint violations demonstrate the effectiveness of our learned flow model.

In Figure 4(d), we show the runtime of the training process. Our approach demonstrates a significantly faster training time compared to NFWPO, with a speed improvement of 2∼3 times. This speed advantage is attributed to the fact that NFWPO requires computationally expensive QP solutions to determine the policy update direction within the feasible action space. In contrast, our approach utilizes the learned normalizing flow model to generate valid actions, with the flow model parameters frozen during the policy update. Although DDPG+P shows a faster runtime, it comes at the expense of a lower average return and higher constraint violations.

## 5   Conclusion

In this work, we present a novel approach called FlowPG based on Normalizing Flows to address action constraints in RL. The architecture of FlowPG allows the policy network to generate actions within the feasible action region. Furthermore, our experimental results demonstrate that FlowPG effectively handles action constraints and outperforms the previous best method by significantly reducing the number of constraint violations while having faster training speeds.

## Acknowledgement

This research/project is supported by the National Research Foundation Singapore and DSO National Laboratories under the AI Singapore Programme (AISG Award No: AISG2- RP-2020-017).

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

## Appendix A    Sample Generation

**HMC:** We used Python implementation [23] of Hamiltonian Monte Carlo algorithm for sample generation of Reacher and Half-Cheetah environments. We set the starting point as the origin($0^D$) and used default hyperparameters such as $\epsilon = 0.2$, $decay = 0.9$.

**PSDD:** In the BSS environment, since there are capacity constraints, restricting each component of the action to the range of $[1, 35]$, we can represent each component of the action as 6-bit integers using Natural Binary Coded Decimal (NBCD). Specifically, we express $a_i, i = 1, \ldots, 5$ as a sum of its binary bits: $a_i = \sum_{j=1}^{6} 2^{6-j} \times a_i^j$, where $a_i^1, \ldots, a_i^6$ are Boolean variables representing each bit of $a_i$. By employing this encoding, we can convert the local constraint $1 \leq a_i \leq 35$ and the global constraint $\sum_{i=1}^{5} a_i = 150$ into Pseudo-Boolean (PB) constraints by substituting each variable with its binary representation.

To compile these PB constraints, we utilize the method described in [6] to convert each constraint into a Sentential Decision Diagram (SDD). The SDDs are then conjoined using the package [24] to create a final SDD. The resulting SDD has a node count of 733, indicating the number of decision nodes, and a model count of 23751, which is the total number of valid actions.

To obtain a Probabilistic Sentential Decision Diagram (PSDD), we parameterize the final SDD. The resulting PSDD has a node count of 802 and a size of 3138. Here, the node count refers to the number of decision nodes in the PSDD, while the size refers to the total number of decompositions.

**Efficiency:** To measure the efficiency of sample generation , we employ a success rate metric, defined as the percentage of valid actions per 100 generated sample points. In both two domains, the HMC method achieves a success rate of 100%. For the rejection sampling, the success rates are 3.93% and 4.7% in the Reacher and Half Cheetah domains, respectively. Figure 5 shows the density of generated sample points within the feasible region. HMC method results in a significantly higher number of data points uniformly distributed across the feasible region. It indicates that HMC is more efficient in sample generation when compared to the rejection sampling method.

When action space constraints are expressed as (in)equalities (such as in the BSS environment), generating valid actions through either rejection sampling or HMC becomes challenging (e.g., rejection/HMC sampling does not produce any action that satisfies all (in)equality constraints within a practical time limit). The advantage of using PSDDs lies in their ability to represent a probability distribution over all valid actions, which implies any sampled action from PSDD is guaranteed to satisfy the constraint. Furthermore, PSDD enables fast sampling of actions with complexity linear in its size and can easily represent uniform distribution over the feasible action space (Section 3.2 in main paper).

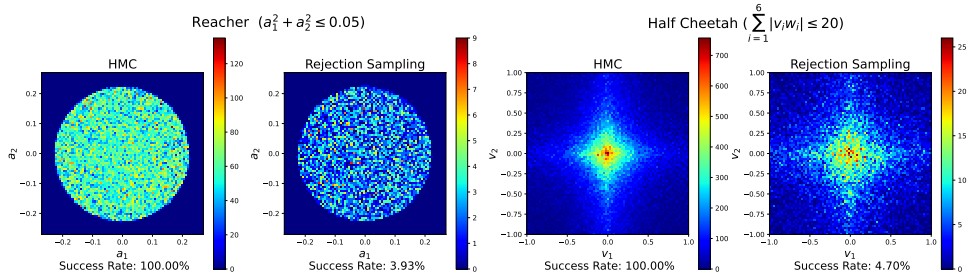

Figure 5: Density map of generated valid actions using HMC and Rejection Sampling methods.

## Appendix B    Training RealNVP

RealNVP [9] was defined using 6 coupling layers as shown in Figure 7. We implemented $k$ and $t$ components as multi-layer perceptrons with 2 hidden layers of size 256 and ReLU activation functions. We used a uniform distribution with range $[-1, 1]^D$ as the latent distribution. For the training purpose, we mollified it with Gaussian Noise of $\mu = 0$ and $\sigma = 0.01$. We used Adam Optimizer [14] for learning the neural network parameters with the learning rate of $10^{-5}$. The trained Flow models were able to produce results mentioned in Table 1. Even when they produce infeasible

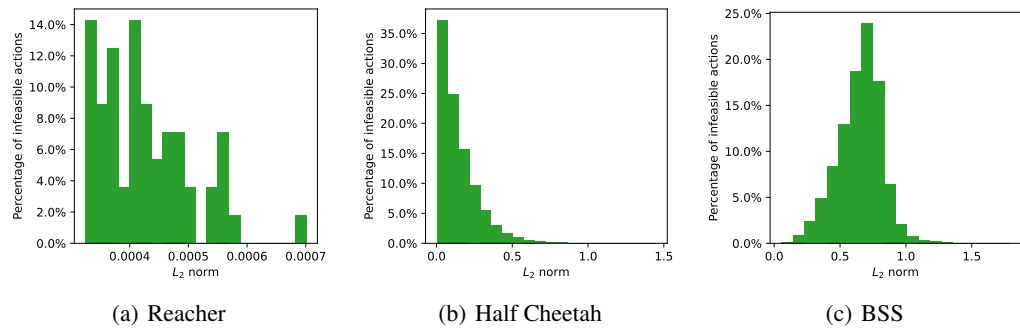

(a) Reacher          (b) Half Cheetah          (c) BSS

Figure 6: $L_2$ distance from the infeasible actions to nearest feasible action.

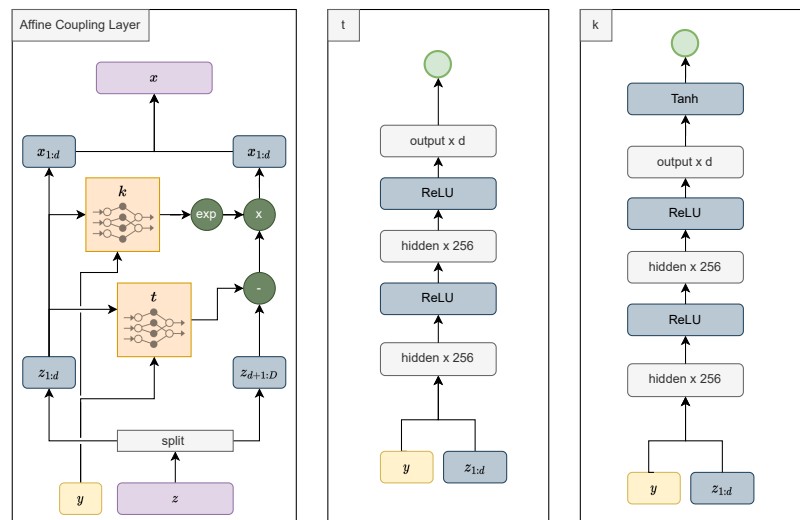

Figure 7: Affine Coupling Layers of RealNVP and $t$, $k$ sub-component networks

| Environment | Accuracy | Recall | Training Time (minutes) |
|---|---|---|---|
| Reacher | 99.98% | 97.85% | $\sim 10$ |
| Half Cheetah | 97.25% | 78.01% | $\sim 120$ |
| Hopper | 87.89% | 81.61% | $\sim 60$ |
| Walker2d | 86.58% | 83.58% | $\sim 120$ |
| BSS | 85.56% | 82.35% | $\sim 60$ |

Table 1: Accuracy, recall of trained flow models and their training time

actions, they tend to be positioned closer to feasible regions. To measure that we filter infeasible actions generated by the trained Flow model and measure the distance ($L_2$ norm) to the nearest feasible action. Figure 6 shows the histogram of the measured $L_2$ norm. This shows that even when our model produces infeasible actions, they are not far away from the feasible region. For example in the Bike sharing environment(BSS) environment, most of the values are between $0$ and $1$, i.e. the error value is no more than a single bike.

## Appendix C    Learning the RL-Model

We defined Actor and Critic networks as 2 hidden layer neural networks with 400 and 300 units respectively as in Figure 8. For the activation function, we used the ReLU activation function. In the final layer of the actor-network, the tanh activation function was used to map output to the $[-1, 1]$ range before passing it to the RealNVP. Adam [14] optimizer was used to train both Actor and Critic

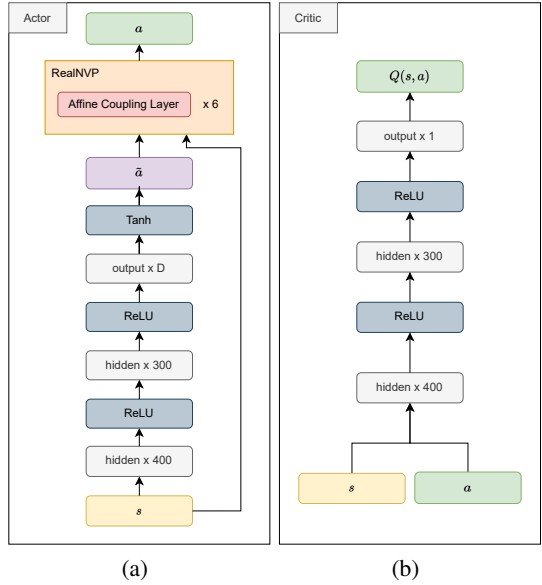

Figure 8: (a) Actor Network and (b) Critic Network

networks with a learning rate of $10^{-4}$ and $10^{-3}$ respectively. For the soft target update, we used $\tau = 0.001$. We used mini-batches of size 64 to train both Actor and Critic networks. We used Gaussian noise with $\mu = 0$ and $\sigma = 0.1$ as action noise for exploration. The replay buffer size was $10^6$. The complete pseudo code of the FlowPG can be found in the Algorithm 1

---

**Algorithm 1** FlowPG Algorithm

---

1: **Input:** Trained $f_\psi$
2: Randomly initialize critic network $Q(s, a; \phi)$ and actor $\mu_\theta(s)$
3: Initialize target network $Q'$ and $\mu'$ with weights $\phi' \leftarrow \phi$, $\theta' \leftarrow \theta$
4: Initialize replay buffer $\mathcal{B}$
5: **for** episode=1,…,$M$ **do**
6:     Initialize the random noise generator $\mathcal{N}$ for action exploration
7:     Reset the environment and retrieve initial state $s_1$
8:     **for** t=1,…,$T$ **do**
9:         Select action $\tilde{a}_t = \mu_\theta(s_t) + \mathcal{N}_t$ based on current policy and exploration noise
10:         Apply flow and get the environment action $a_t = f_\psi(\tilde{a}_t, s_t)$
11:         **if** $a_t$ is invalid **then**
12:             $a_t \leftarrow QP\_Solver(a_t, s_t, \mathcal{C}(s_t))$
13:         **end if**
14:         Execute action $a_t$ and observe reward $r_t$ and next state $s_{t+1}$
15:         Store transition $(s_t, a_t, r_t, s_{t+1})$ in $\mathcal{B}$
16:         Sample a random minibatch of $N$ transitions $(s_i, a_i, r_i, s_{i+1})$ from $\mathcal{B}$
17:         $y_i = r_i + \gamma Q'(s_{i+1}, f_\psi(\mu_{\theta'}(s_{i+1}), s_{i+1}); \phi')$
18:         Update critic by minimizing the loss: $\mathcal{L} = \frac{1}{N} \sum_i (y_i - Q(s_i, a_i; \phi))^2$
19:         Update actor policy using the sampled policy gradient:
20:             $\nabla_\theta J(\mu_\theta) = \frac{1}{N} \sum_i \nabla_a Q(s_i, a; \phi) \nabla_{\tilde{a}} f_\psi(\tilde{a}, s_i) \nabla_\theta \mu_\theta(s_i)|_{\tilde{a}=\mu_\theta(s_i), a=f_\psi(\tilde{a}, s_i)}$
21:         Update target networks:
22:             $\phi' \leftarrow \tau\phi + (1 - \tau)\phi'$
23:             $\theta' \leftarrow \tau\theta + (1 - \tau)\theta'$
24:     **end for**
25: **end for**

---

