# OpenReview forum: "FlowPG: Action-constrained Policy Gradient with Normalizing Flows"
_NeurIPS.cc/2023/Conference — NeurIPS 2023 poster_

### Official Review · Reviewer_Vb4Z · 2023-06-29

**Soundness:** 3 good
**Presentation:** 3 good
**Contribution:** 3 good
**Rating:** 6
**Confidence:** 4

**Summary:**

Handling constraints in reinforcement learning is a fundamental problem that has application in areas such as robotics and resource allocation. A common solution is to incorporate a projection step to compute feasible actions, which involves a computationally expensive optimization solver in the loop. This can be prohibitively slow, especially when constraints are non-convex. When used as part of the policy with a differentiable solver, it can led to the zero gradient problem if the policy output is far outside the feasible space. To circumvent the need for an optimization solver, the authors propose to learn an action mapping which respects the constraints with high probability by design. Specifically, they train a normalizing flow offline on samples from the distribution of feasible actions. These samples are generated via Hamiltonian Monte-Carlo for continuous action spaces and probabilistic sentential decision diagrams for discrete action spaces. During RL, the policy then outputs a latent action, which is mapped to the space of feasible actions via the normalizing flow. The weights of the flow are frozen, but the gradients are still propagated through the network to train the policy. They evaluate their approach, FlowPG, on two continuous control robotics tasks and a discrete resource allocation problem. FlowPG achieves the highest average return across ten random seeds for all problems and has the lowest number of constraint violations (prior to a projection step). When it does violate a constraint, the magnitude of the violation is lower, meaning it is closer to the feasible action space. And it achieves these benefits with reduced wall clock time compared to the best baseline.

**Strengths:**

- Addressing action constraints in reinforcement learning is an important problem and critical for many real-world robotics and decision making tasks.
- FlowPG is a novel solution to this problem and effective in improving performance while reducing constraint violations and wall clock time, all while being fairly straightforward to implement.
- The finding that a uniform prior reduces constraint violations when used in conjunction with RL is a useful insight. The bounded support of the uniform distribution also works nicely with the fact that policy gradient algorithms often work well when the output is passed through a squashing function to respect box constraints on action limits.
- Constraint violations which do occur, however infrequently, can still be remedied by solving an optimization problem. Importantly, this does not need to be done when performing the policy updates, just during rollouts, which makes training more efficient.
- The paper is well organized and clearly written. It does a good job explaining the novelty and results and provides enough information to support its claims. In particular, I liked the visualizations in Figure 3 of the learned action space for the Reacher task.

**Weaknesses:**

- The tasks considered in the paper are fairly standard benchmarks and showcase the effectiveness of FlowPG. However, the constraints in each task are relatively simple, with only Half Cheetah having constraints which depend on a portion of state. This makes it difficult to gauge how this approach will scale to more challenging, state-dependent constraints. Tasks with more complex constraints would significantly strengthen the paper.
- It seems difficult to scale the sample generation procedure to higher-dimensional action spaces. And for state-dependent constraints, it seems challenging to span the relevant portions of the state space which will be visited by the intermediate and final policies used during learning. Again, more challenging tasks would help prove the concept. It may be that an iterative approach which refines the flow on relevant portions of the state space will be necessary.
- There does not appear to be any discussion of how the discrete actions in the BSS environment are handled. My guess is that the flow is trained on the integer values, but the output will still be continuous. We can then just round the output, but this could result in constraint violations. A better discussion of how this is handled in the main paper would help.
- There is a lengthly discussion of gradient computation for the normalizing flow, but this is usually just handled by the deep learning framework. It seems unnecessary to get into these details unless the form of these gradients is leveraged to improve the speed of training. If auto-differentiation is still used as normal, then this feels a bit like filler and could be replaced with more relevant details about training, the tasks, or results.
- The motivation for using a normalizing flow, rather than other generative models, seems missing in the paper. The authors do mention something, but it felt a bit hand-wavy. If we were optimizing a stochastic policy with an on-policy algorithm, such as PPO, having a tractable likelihood would be really important. This would be a great motivation for using a normalizing flow. But since we are using a deterministic policy and training with DDPG, it seems more arbitrary.

**Questions:**

- How do you think this approach would scale to more complex, state-dependent constraints? Would it be too difficult to generate samples which cover the state space sufficiently? And if so, would an iterative approach which refines the flow on states encountered under the current policy be feasible?
- How are the discrete actions in the BSS environment handled? If the output of the flow is rounded, does this lead to more constraint violations?
- Is the form of the normalizing flow leveraged to more efficiently compute gradients while training? Or are we using standard auto-differentiation?
- What is the rationale for using a normalizing flow over other generative models in the deterministic policy case?
- How do you think this approach would work for an on-policy algorithm, such as PPO, which requires the log-likelihood?
- How was the "hard wall" for handling constraints implemented with HMC, which needs a differentiable log-likelihood? Were they just barrier functions?

**Limitations:**

The authors discuss how their approach can still lead to constraint violations, requiring the use of an optimization solver in this case. However, they show that the probability of constraint violation is significantly reduced. A possible limitation not really discussed is scalability due to the need for sample generation to train the flow. It may be hard to scale this approach to higher-dimensional action spaces and constraints which heavily depend on the state. The Half Cheetah task definitely shows it can work, so at the very least, this is a good preliminary study to illustrate the potential of the approach.

---

> ### Author Rebuttal · Authors · 2023-08-10
>
> 1. How to generate samples to cover the state space sufficiently? Refine flow during training?
>
> Indeed, this is a good point. We note that our approach models the space of feasible actions for different states. Therefore, even if the feasible state space is large, but the relationship between different states and feasible action space is roughly similar, our approach can work well.
>
> If the space of feasible actions varies significantly for different states, then training flow in a static manner would be challenging. Refining the flow during the training so that the flow can focus on reachable part of the state space, similar to training of the Q function using replay buffer, is indeed a good idea. Our overall approach can be easily modified to perform this online flow training. We shall explore it actively for larger problems with more complex state-action constraints. In our current settings, we were able to get good accuracy and recall without this online flow training as noted in table 1 in rebuttal pdf.
>
>
> 2. How to handle discrete action?
>
> The simulator for BSS is equipped with a rounding procedure that tries to apply a heuristic method to extract an integer solution from a given action. In our experience, this method was successful for most cases. When this rounding method did not give an integer solution satisfying all the constraints, we solved an integer program to perform the projection. This approach is applied consistently to all the baselines.
>
> 3. Is the form of the normalizing flow leveraged to more efficiently compute gradients while training? Or are we using standard auto-differentiation?
>
> We use Auto-differentiation. However, we wanted to explicitly highlight how the policy update in Eq 11 (main paper) utilizes the gradient from the flow model. In particular, in Eq 13, 14, highlight the policy gradient part coming from the flow.
>
>
> 4. Rationale for using normalizing flows
>
> Please see the common response. (Sec. 1)
>
> 5. PPO+Flow
>
> Assuming the trained flow model achieves a high accuracy rate and recall rate, the outputs of the flow model are likely to be valid actions. In this case, the PPO approach can be effectively employed. This is due to the fact that the log probabilities of actions can be computed using the flow and that the optimization of policy is conducted within the feasible region.
>
> However, in cases where projection is required and the projected action is mapped using the inverse transformation of the flow to an area beyond the domain of the latent variable (e.g., outside $[-1, 1]$ for uniform prior), then we need to compute the closest point in the latent space and use this as a proxy to compute the log-probability of the action.
>
>
> 6. How was the "hard wall" for handling constraints implemented with HMC, which needs a differentiable log-likelihood? Were they just barrier functions?
>
> We initially tried with differentiable barrier functions. However, we found that better or comparable results could be achieved through a non-differentiable function with sharp barriers. Specifically, we employed a piecewise function that takes on a value of $-\infty$ if there is constraint violation, and $0$ otherwise. We set the gradient to be zero everywhere.
> For further details and precise implementation, you may refer to the `sample_generation/hmc.py` file in the supplementary materials.

---

> > ### Comment · Reviewer_Vb4Z · 2023-08-14
> >
> > > Rationale for using normalizing flows
> >
> > The empirical results provided in the common rebuttal response are convincing and should be included in the final paper. If placed in the appendix, there should be a reference to it in the main paper.
> >
> > > Other action-constrained RL domains (from common response)
> >
> > Thank you for these additional results, they are definitely promising! What are the state features used in Hopper and Half Cheetah? Are they also velocities?
> >
> > > PPO + Flow
> >
> > That makes sense! An additional question, though, is that when using DDPG, you do not project during the policy update, right? Could you not also do this in the case of PPO? The projection, if needed, could be considered part of the environment, and the action added to the replay buffer is the one output by the flow. However, I could see how this may reinforce bad actions and lead to more constraint violations.

---

> > > ### Author Response · Authors · 2023-08-15
> > >
> > > Thank you very much for the response and comments. We shall include these new experimental results in the next revised version.
> > >
> > > > State features used in Hopper and Half Cheetah
> > >
> > > Yes, the reviewer is correct. The state features involved in the constraints for Half-Cheetah, Hopper, and Walker2d are angular velocities.
> > >
> > > > When using DDPG, you do not project during the policy update, right?
> > >
> > > We do not project action during the policy update in DDPG.
> > >
> > > > Could you not also do this in the case of PPO?
> > >
> > > We agree that storing the actions generated by the flow model (pre-projected action) is possible for PPO. We note that when the trained flow model achieves high accuracy and recall rates, even if a pre-projected action violates the constraints, the distance between the pre-projected action and the projected action tends to be relatively small (as shown in Column 3, Figure 4 in main paper). In such cases, the constraint violations might still remain within reasonable limits, and the zero gradient problem would not be a major issue.
> > >
> > > However, if the distance between projected and pre-projected actions is significant (e.g., when the flow is not trained well), then as the reviewer also mentions, it may lead to higher constraint violations and the zero gradient problem.

---

### Official Review · Reviewer_JVCX · 2023-07-02

**Soundness:** 3 good
**Presentation:** 3 good
**Contribution:** 3 good
**Rating:** 5
**Confidence:** 3

**Summary:**

This paper solves the problem of action-constraint reinforcement learning. The author utilizes the Flow model to learn a projection from the action to the latent variable, and then integrate DDPG to construct the FlowPG framework. Empirically FlowPG outperforms its competitors in both fewer constraint violations and faster in elapsed time.

**Strengths:**

1. The manuscript is clearly structured and well presented, and it is easy to read.
2. It seems interesting to apply the generative model in constraints optimization problem. Introducing the flow model is novel in action-constraint scenarios, and it effectively avoids solving a QP problem after the policy network.
3. The authors utilizes HMC and PSDD in sampling from the valid actions set.

**Weaknesses:**

The action-constraint problems is actually somehow similar with constraint-RL problem, and there are some important works such as [1,2] and other related works. It is suggested that the authors should provide a discussion about whether other constraint RL methods could be applied in the action-constraint scenarios, and provide some experimental result if possible.

[1] Constrained Policy Optimization https://arxiv.org/pdf/1705.10528.pdf

[2] Safety-Constrained Reinforcement Learning for MDPs. https://arxiv.org/abs/1510.05880

Another issue is that the action-constraint RL problem could be seen as offline RL problem (invalid actions space could be OOD actions). There are various offline RL methods appearing [3,4] in recent years ([5] also applies the flow model), and the authors are suggested to consider the offline RL methods in this manuscript.

[3] Off-Policy Deep Reinforcement Learning without Exploration. https://arxiv.org/abs/1812.02900

[4] Stabilizing off- policy q-learning via bootstrapping error reduction. https://arxiv.org/abs/1906.00949

[5] APAC: Authorized Probability-controlled Actor-Critic For Offline Reinforcement Learning. https://arxiv.org/abs/2301.12130

**Questions:**

1. There are some confusions in generating valid action space, even with HMC and PSDD, could it be assured that the $\tilde{a}$ could be transformed to a valid action after the flow model projection in Figure 2(a). In addition, as we want to maximize the cumulative reward in RL problem, should the input of the flow model be ($\tilde{a}, s, s'$) where $s'$ is the next state. It is also noted when $\tilde{a}$ becomes a valid action $a$, the next state $s'$ may also changes

2. The author claims that flow model is more efficient than VAE and GAN for data generation. Could the author provide some ablation study on this issue, especially for generating valid action samples.

3. The experiments seem not convincing enough. The authors only considers Half-Cheetah and Bike Sharing System. Actually Half-Cheetah belongs to D4RL tasks, and there are many other tasks inside (some could be constraint RL tasks), it is suggested to consider more dataset for comparisons.

4. In addition, the MCMC style methods are often time-consuming, how is the entire training time, compared with other competitors?

**Limitations:**

see weakness and questions

---

> ### Author Rebuttal · Authors · 2023-08-10
>
> 1. Other constrained RL methods
>
> In our paper, we consider the scenario where constraints are imposed on actions at each RL step, and these constraints have closed forms. Unlike the standard constrained MDP, we do not define cost functions for individual state-action pairs. Therefore, the direct application of well-known techniques like CPO[1] and Lagrangian Relaxation[2] becomes challenging for solving action-constrained RL. Additionally, these approaches do not provide a guarantee of constraint satisfaction. We shall clarify this point in the revision.
>
> 2. Connection to Offline RL methods
>
> Thanks for pointing to this line of work, we shall certainly discuss it in the revision. We do highlight some key differences from off-policy RL methods:
>
> – Off policy RL methods require a dataset for training from an expert or random policies. This is not required in our method as action constraints are known, and we use HMC/PSDD to sample feasible actions for different states.
>
> – Notice that collecting off-policy data from a random policy for offline RL is generally trivial. However, this is not trivial for our action constrained setting. Even a random policy must select actions uniformly from *only* the feasible action space, which itself is a key component we are trying to address.
>
>
> 3. Input to flow? Is output of the flow guaranteed to be valid?
>
> We shall clarify that the actions generated by the trained flow model are not guaranteed to be valid actions. However, a well-trained flow model exhibiting a high accuracy rate, such as 99.98% in the Reacher domain, is highly likely to generate valid actions. Even in instances where an invalid action is generated, our experiments show that these invalid actions are generally closer to the feasible region when compared to actions generated by the DDPG+P and NFWPO approaches. We also note that the projection of invalid actions into the feasible region is required in all approaches since the environment only accepts valid actions.
>
> The input to the normalizing flows model are $s$ and $\tilde{a}$, where $\tilde{a}$ belongs to the latent distribution of the flow model. This $\tilde{a}$ is transformed into the environment action $a$ by the flow model, which is then fed into the RL simulator (after projection, if required). The dynamics of the environment (i.e., the next state) only depends on $s$ and $a$. The input to the flow cannot include the next state s’ as the policy output $\tilde{a}$ cannot be executed by the environment directly before mapping by the flow model. Thus s’ is not available as an input for the flow model.
>
> Reward maximization is primarily handled by the policy gradient part connected with the flow model as shown in Figure 2a in the paper.
>
> 4. Other generative models such as VAE and GAN
>
> Please see the common response. (Sec. 1)
>
> 5.  More domains in D4RL tasks
>
> Please see the common response. (Sec. 3)
>
> 6. Entire training time when considering HMC data generation
>
> In our approach, generating valid actions and training a normalizing flows model are conducted prior to RL training. Specifically, the generation of data using HMC takes less than 5 minutes to produce 1 million samples for all the domains.This rapid data generation is attributed to the efficiency of HMC. An ablation study comparing the efficiency of HMC and traditional rejection sampling has been included in the general response.
>
> The training time for the flow model for different domains is shown in Table 2 of rebuttal PDF. We do highlight that in our approach, once the flow is trained, retraining is not required if some problem parameters change such as the initial state, reward function etc. However, for approaches such as NFWPO, every time we run the approach, projection steps will be required, thus leading to higher runtime (as shown in figure 4, last column in main paper).

---

> > ### Comment · Reviewer_JVCX · 2023-08-16
> > **Thank you for your rebuttal**
> >
> > Thank you for your detailed reply. Your response has addressed some of my concerns, and I do appreciate your work.
> >
> >   **Other generative models such as VAE and GAN**
> > The section 1 in common response is our strong enough to convince me, and comparisons between flow model, wgan and vae are somehow weak.  More empirical studies are suggested.
> >
> > **Entire training time when considering HMC data generation**
> > Thank you for supplemental studies. I am OK with about the training time study.

---

> > > ### Author Response · Authors · 2023-08-21
> > >
> > > Thank you for your response. As noted in the rebuttal pdf, just measuring accuracy alone for a model can be misleading due to potentially low recall rate (i.e., limited coverage of the feasible action space). Low recall rate implies that the RL policy does not optimize over the entire feasible region.
> > >
> > > As we noted in our common rebuttal (under “Rationale for using normalizing flows”), it is not straightforward to compute the recall rate for WGAN and VAE as these are not invertible models, unlike the normalizing flows. Nonetheless, we can use the Wasserstein distance (W-Dist) as a proxy to approximate the recall rate for non-invertible models. We follow the steps below:
> > >
> > > 1. Generate data points using HMC (i.e., from the feasible action space)
> > > 2. Generate data points using generative models such as Flow/WGAN
> > > 3. Compute the Wasserstein distance between two datasets generated in Step 1 and 2 using the paper (and its publicly available GitHub implementation “​​geomloss”) [5]
> > >
> > > If the W-Dist is low (close to zero), then it means the difference between two distributions is small. That is, in our case, recall rate of the model used in Step 2 is high (which is desirable) as the data generated from the model is close to the feasible actions generated from the HMC sampling. If the W-Dist is high, it analogously represents a low recall rate. Our results for different domains are as follows (we generated 100K samples in Step 1 and 2 each):
> > >
> > >
> > > ```
> > > +--------------+----------+----------+
> > > | Problem      |   WGAN   |   Flow   |
> > > +==============+==========+==========+
> > > | Reacher      | 0.000058 | 0.000002 |
> > > +--------------+----------+----------+
> > > | Half-Cheetah | 1.181345 | 0.096802 |
> > > +--------------+----------+----------+
> > > | Hopper       | 0.315453 | 0.009221 |
> > > +--------------+----------+----------+
> > > | Walker2D     | 0.302288 | 0.088734 |
> > > +--------------+----------+----------+
> > >
> > > ```
> > > The W-Dist in the Flow model is significantly smaller than the distance in WGAN by more than an order of magnitude in most cases. This indicates a significantly high recall rate for the Flow model. Despite tuning WGAN's hyperparameters over multiple days, we were not able to further improve the WGAN’s results.
> > >
> > > We also point the reviewer to paper: An Empirical Comparison of GANs and Normalizing Flows for Density Estimation, 2021 [3]. This paper also shows that the Flow models outperform WGAN on several different types of distributions.
> > >
> > > The VAE model was much worse than WGAN (see Figure 3b in rebuttal PDF). Therefore, given the limited time, we did not explore VAE further.
> > >
> > > [3] Liu, Tianci, and Jeffrey Regier. "An Empirical Comparison of GANs and Normalizing Flows for Density Estimation." arXiv preprint arXiv:2006.10175 (2020).
> > >
> > > [5] Feydy, Jean, et al. "Interpolating between optimal transport and MMD using sinkhorn divergences." The 22nd International Conference on Artificial Intelligence and Statistics. PMLR, 2019.

---

### Official Review · Reviewer_RnL4 · 2023-07-03

**Soundness:** 3 good
**Presentation:** 3 good
**Contribution:** 3 good
**Rating:** 5
**Confidence:** 2

**Summary:**

This paper provide a new method for ACRL, which incorporate normalized flow methods to alleviate the action violation problem. It achieves better results on MuJuCo compared with other methods.

**Strengths:**

- Introduce normalized flow into action control -- which maps the original hard-to-control action space into another easy-to-control space.
- Design a HMC-PSDD framework to efficiently train the flow model.
- Find an appropriate prior distribution for the flow model.

**Weaknesses:**

Main issue:
- An ablation study for PSDD may be necessary -- as we don't know the quality of generated valid actions. For example, a possible baseline can be: interact with the environments many times to get the possible valid actions.


Minor issue:
- DDPG is relatively outdated. More recent SoTA methods should be added into the experiments.

**Questions:**

I am not an expert in ACRL so I wonder if there are other ACRL benchmarks? Only 3 environments seem not enough.

**Limitations:**

No obvious negative societal impacts.

---

> ### Author Rebuttal · Authors · 2023-08-10
>
> 1. Ablation study for PSDD and HMC
>
> Please see the common response.  (Sec. 2)
>
> 2. Comparison with recent ACRL algorithms + more domains
>
> Please see the common response.  (Sec. 3 and 4)

---

> ### Comment · Reviewer_RnL4 · 2023-08-14
>
> Thanks for the detailed explanation!

---

> > ### Author Response · Authors · 2023-08-15
> >
> > Thank you very much for reviwing our response. We hope that we have addressed your concerns, and we shall include the ablation study and additional experiments in our revised version.

---

### Official Review · Reviewer_NbGB · 2023-07-05

**Soundness:** 3 good
**Presentation:** 3 good
**Contribution:** 2 fair
**Rating:** 5
**Confidence:** 3

**Summary:**

The paper introduces a novel action-constrained reinforcement learning (ACRL) algorithm called FlowPG, which utilizes a normalizing flow model to generate actions within the feasible action region. Experimental results demonstrate that FlowPG effectively handles action constraints and outperforms two existing ACRL algorithms by reducing the number of constraint violations.

**Strengths:**

The paper is well-written and provides good motivation.

**Weaknesses:**

The paper lacks the ablation study to validate importance of the proposed HMC and PSDD. The advantages of training speeds are not obvious.

**Questions:**

Q1. The motivation behind the use of Hamiltonian Monte Carlo (HMC) and probabilistic sentential decision diagrams (PSDD) is not sufficiently clear. Additionally, an ablation study is necessary to validate the importance of these proposed techniques.

Q2. The paper claims that the proposed method achieves faster training speeds compared to the baselines. However, Figure 4(a) shows no clear advantages in terms of convergence speed for FlowPG compared to the two baselines. This discrepancy needs to be addressed and clarified.

Q3. The paper would benefit from including more comparisons between the proposed method and recent ACRL algorithms. By providing such comparisons, the authors can further highlight the strengths and weaknesses of their approach and provide a more comprehensive evaluation.


**Limitations:**

see Questions

---

> ### Author Rebuttal · Authors · 2023-08-10
>
> 1. Motivation behind HMC and PSDD. Ablation study to justify the reason of using them
>
> Please see the common response. (Sec. 2)
>
> 2. Convergence speed of FlowPG
>
> For half cheetah and reacher, we shall highlight that our approach has a similar training curve as NFWPO. However, the key difference is that each training step is much faster in our approach than NFWPO as shown in Figure 4 (last column). Therefore, in terms of runtime, our approach converges much faster than NFWPO. For BSS domain, our approach has better training curve than NFWPO (in terms of # of training steps), and is also significantly faster as shown in Figure 4 (first and last column).
>
> 3. Comparison with recent ACRL algorithms
>
> Please see the common response. (Sec. 4)
>
> Citations for all our responses:
>
> [1] Achiam, Joshua, et al. "Constrained policy optimization." International conference on machine learning. PMLR, 2017.
>
> [2] Tessler, Chen, Daniel J. Mankowitz, and Shie Mannor. "Reward constrained policy optimization." arXiv preprint arXiv:1805.11074 (2018).
>
> [3] Liu, Tianci, and Jeffrey Regier. "An Empirical Comparison of GANs and Normalizing Flows for Density Estimation." arXiv preprint arXiv:2006.10175 (2020).
>
> [4] Lin, Jyun-Li, et al. "Escaping from zero gradient: Revisiting action-constrained reinforcement learning via Frank-Wolfe policy optimization." Uncertainty in Artificial Intelligence. PMLR, 2021.

---

> > ### Comment · Reviewer_NbGB · 2023-08-17
> > **Response**
> >
> > Thanks for the additional experiments and explanation, that address my concerns.

---

> > > ### Author Response · Authors · 2023-08-22
> > >
> > > Thank you very much for reviewing our response. We shall include the ablation study and additional experiments in our revised version.

---

### Author Rebuttal · Authors · 2023-08-10

We thank all the reviewers for their thoughtful feedback and suggestions. We would like to address a few common questions as follows.

1. Rationale for using normalizing flows

Our reasons are:

High Accuracy: We conducted an ablation study comparing different generative models such as VAE and WGAN (which is more stable than GAN) in the Reacher domain with the constraint $a_1^2 + a_2^2 \leq 0.05$. We evaluated the accuracy by calculating the percentage of valid actions among 100k generated actions via such models respectively. The accuracy of the various generative models was as follows: Normalizing Flow: 99.98%; WGAN: 98%; VAE: 83%.

Recall Through Invertible Transformations: Normalizing flows provide the ability to measure recall due to their invertible bijective transformations. The recall rate (we also call it “coverage”) indicates the fraction of valid actions that can be generated from the latent space. Given a state $s$, it can be computed as follows $recall(s) = \frac{\sum_{a\in\tilde{\mathcal{C}}(s)} \mathbb{I}\_{\textbf{dom}f_{\psi}}\big(f^{-1}\_{\psi}(a,s) \big) }{|\tilde{\mathcal{C}}(s)|}$

Where $\tilde{\mathcal{C}}(s)$ is the set of valid actions that are uniformly distributed in feasible region. $f^{-1}\_{\psi}$ is the inverse transformation function of the normalizing flows model $f\_\psi$. We compute the average recall over all uniformly sampled states to obtain the recall of our flow model. The achieved recall rates for our trained normalizing flow model are as follows: 97.85% for the Reacher, 78.01% for the Half Cheetah, and 82.35% for the BSS environment. In contrast, recall rate cannot be computed in a straightforward fashion in VAE and WGAN since determining the corresponding latent action for a given valid action is not possible [3].  Nonetheless, we still can visualize the coverage of VAE and WGAN in Reacher. In Figure 3 in the rebuttal PDF, we can see that the feasible region is not fully covered in both VAE and WGAN models.

A higher accuracy rate indicates fewer projection operations. A higher recall rate indicates that the agent is able to explore a nearly complete feasible region. Table 1 in the rebuttal PDF summarizes the accuracy and recall


2. Ablation study on HMC and PSDD

To measure the efficiency of sample generation , we employ a success rate metric, defined as the percentage of valid actions per 100 generated sample points. In both two domains, the HMC method achieves a success rate of 100%. For the rejection sampling, the success rates are 3.93% and 4.7% in the Reacher and Half Cheetah domains, respectively. Figure 1 in the rebuttal PDF shows the density of generated sample points within the feasible region. HMC method results in a significantly higher number of data points uniformly distributed across the feasible region. It indicates that HMC is more efficient in sample generation when compared to the rejection sampling method.

When action space constraints are expressed as (in)equalities  (such as in the BSS environment), generating valid actions through either rejection sampling or HMC becomes challenging (e.g., rejection/HMC sampling does not produce any action that satisfies all (in)equality constraints within a practical time limit). The advantage of using PSDDs lies in their ability to represent a probability distribution over all valid actions, which implies any sampled action from PSDD is guaranteed to satisfy the constraint. Furthermore, PSDD enables fast sampling of actions with complexity linear in its size and can easily represent uniform distribution over the feasible action space (section 3.2 in main paper).

3. Other action-constrained RL domains

We have evaluated our proposed framework in two more domains Hopper and Walker, which belong to the Gym-MuJoCo continuous control task. In Hopper domain, we consider the state-dependent constraint $\sum\_{i=1}^3 \text{max}(w\_i a\_i, 0) \leq 10$ , where $w\_i$ is the state feature. This is similar to the constraint in Half Cheetah.Similarly, we consider constraint $\sum\_{i=1}^6 \text{max}(w\_i a\_i, 0)\leq 10$ in Walker domain. All experimental settings remain the same as in the paper.

Figure 2 in the rebuttal PDF shows different curves during training in Hopper and Walker. In Hopper domain, when comparing to DDPG+P, our approach has less constraint violations before the projection operation, and achieves comparable results in terms of average return, magnitude of constraint violation, and running speed. In contrast to the NFWPO approach, our method performs better across all metrics, except for minor differences in the magnitude of constraint violation. In Walker domain, our approach outperforms DDPG+P across all metrics except running time. When compared with NFWPO, while our approach has slightly more cumulative constraint violations, the achieved magnitude of constraint violation and average return remain comparable. Significantly, our approach has much faster runtime compared to NFWPO.

To summarize, in the two new domains featuring state-dependent constraints, our approach consistently achieves a comparable or better average returns as other baselines. Moreover, our approach also demonstrates significantly reduced constraint violations, faster running time in comparison to other baselines.

4. Comparison with recent ACRL algorithms

In  constrained RL, most works focus on tackling cumulative constraints such as $\mathbb{E}\_{\pi}[\sum_{t=0}^\infty C(s\_t,a\_t)] \leq c$. There exist a limited number of works that specifically target ACRL where constraints are expressed as closed-form conditions on actions. Among the existing solutions for ACRL, representative approaches include DDPG+Projection, SAC+Projection, DDPG+OptLayer, and NFWPO. Among these, NFWPO has been demonstrated to have better performance in terms of return and constraint violations before projection [4]. This is why we have selected NFWPO as our primary baseline for comparison.

---

### Decision · Program_Chairs · 2023-09-21

**Decision:**

Accept (poster)

**Comment:**

The reviews and discussion offered enough positive support, with some concerns being good topics for future work. We encourage the authors to use the feedback to revise and strengthen the camera-ready.